



# Review article: Using spaceborne lidar for snow depth retrievals: Recent findings and utility for global hydrologic applications

Zachary Fair[1,2], Carrie Vuyovich[1], Thomas Neumann[3], Justin Pflug[1,2], David Shean[4], Ellyn M. Enderlin[5], Karina Zikan[5], Hannah Besso[4], Jessica Lundquist[4], Cesar Deschamps-Berger[6], and Désirée Treichler[7]

[1]Hydrological Sciences Laboratory, NASA Goddard Space Flight Center, Greenbelt, MD, USA
[2]Earth System Science Interdisciplinary Center, University of Maryland, College Park, MD, USA
[3]Earth Sciences Division, NASA Goddard Space Flight Center, Greenbelt, MD, USA
[4]University of Washington, Seattle, WA, USA
[5]Department of Geosciences, Boise State University, Boise, ID, USA
[6]Pyrenean Institute of Ecology-CSIC, Avda Montañana 1005, Zaragoza 50.059. Spain
[7]Department of Geosciences, University of Oslo, Oslo, Norway

**Correspondence:** Zachary Fair (zachary.fair@nasa.gov)

**Abstract.** Lidar is an effective tool to measure snow depth over key watersheds across the United States. Lidar-derived snow depth observations from airborne platforms have demonstrated centimeter-level accuracy at high spatial resolution. However, ground-based and airborne lidar surveys are costly and limited in space and time. In recent years, there has been an emerging interest in using spaceborne lidar to estimate snow depth. Preliminary results from spaceborne lidar altimeters such as

5  the NASA Ice, Cloud, and Land Elevation Satellite-2 (ICESat-2) can provide routine snow depth retrievals over watersheds, though further research on accuracy, coverage, and operational potential is needed. In this review, we outline the current status of research using spaceborne lidar to derive snow depth. We focus on the currently operational ICESat-2 mission, with a summary of snow observations gathered from previous studies. We also outline best practices for spaceborne lidar snow depth retrieval, based on findings from recent studies. We conclude with a discussion of ongoing challenges for spaceborne lidar,

10  with suggestions for future studies and requirements for future mission concepts.



# 1 Introduction

Seasonal snow is a critical factor in Earth's climatological, ecological, and hydrological processes. Annually, seasonal snow covers a maximum extent of approximately 36% of the Earth's Northern Hemisphere (Estilow et al., 2015; Wrzesien et al., 2019), reflecting a significant portion of the incoming solar radiation and helping to cool the planet. Snow plays an integral role in the well-being of many high-latitude wildlife species and ecosystems, including the boreal forest, the largest terrestrial ecosystem (Boelman et al., 2019; Reinking et al., 2022). Melt water from seasonal snow accounts for approximately one-sixth of the world's freshwater supply and supports numerous hydrologic applications including hydropower, agriculture, and recreation (Barnett et al., 2005; Li et al., 2017). For these reasons, snow is listed as a future research need in a recent report by the Intergovernmental Panel on Climate Change (IPCC, 2022). Snow water equivalent (SWE) and depth are also identified as Essential Climate Variables needed to better understand our changing climate by the Global Climate Observing System Implementation Plan (GCOS, 2022) and the 2017 Decadal Survey for earth science (NASEM, 2018).

As snow is highly variable over space and time (Sturm et al., 2010), it is especially important to capture SWE heterogeneity at basin-wide scales to accurately reproduce observed snowmelt, runoff, and streamflow in models (Brauchli et al., 2017; DeBeer and Pomeroy, 2017; Kiewiet et al., 2022). Frequent, high-resolution, spatially-distributed observations are needed to characterize this important component of the water and energy cycle. The observational requirements for SWE and snow depth stated in the GCOS Implementation Plan and the 2017 Decadal Survey suggest a spatial resolution of 500 m to 1 km, with higher resolution (100 m) needed in the mountains. Additional requirements include a temporal frequency of 1-5 days and an accuracy of 10-20%. While many properties of snow are currently observable globally by satellites, including snow extent and albedo, we currently lack information about snow depth and SWE at the recommended scales needed to inform climate and water resource applications.

Recent studies have shown that it is possible to measure snow depth and fill gaps in global snow measurement capabilities from space using lidar altimetry. This is an appealing alternative to in-situ and airborne lidar methods because of its potential for global-scale observations. Spaceborne lidar derives snow depth using methods established with airborne lidar, including differential altimetry, and unique methods (Section 4). Spaceborne lidar altimeters, such as the Ice, Cloud, and Land Elevation Satellite-2 (ICESat-2; Markus et al., 2017) and the Global Ecosystem Dynamics Investigation (GEDI; Dubayah et al., 2020), provide accurate elevation observations, with centimeter-level accuracy for ICESat-2 over ice sheets and decimeter-level accuracy for both GEDI and ICESat-2 over forests (Adam et al., 2020; Brunt et al., 2021). Although current spaceborne platforms have relatively long revisit times and coarse across-track sampling, satellite altimetry could theoretically be used for routine measurements of snow depth over key watersheds.

In this paper, we review the current status of research using spaceborne lidar, and evaluate its potential to derive snow depth to meet the research and operational needs of the hydrology community. Our discussion concentrates on the currently operational ICESat-2 mission. We summarize published studies and present a case study over the tundra of Alaska to demonstrate accuracy and uncertainty estimates for several current methods. We also document challenges for current measurement approaches, with suggestions for future studies. We focus on terrestrial snow in this paper, but we acknowledge that snow depth retrievals have





also been attempted over land ice (Enderlin et al., 2022; Hu et al., 2022b; Lu et al., 2022) and sea ice (Hu et al., 2022b; Kwok et al., 2020; Lu et al., 2022). Snow depth measurements over ice masses have different challenges that are outside the scope of this review.



## 2 Basic Lidar Principles

Current space-based lidar instruments used to measure snow depth have two primary measurement modalities: waveform-
based and photon-counting. Waveform lidar systems record the change in amplitude, or signal strength of the return, over
time. The shape of the received waveform is sensitive to terrain characteristics such as surface roughness, which may cause
centimeter-to-decimeter levels of bias in the final elevation measurement (Dong and Chen, 2017). Waveform-based spaceborne
lidars include the original ICESat (Schutz et al., 2005) and the GEDI instrument (Dubayah et al., 2020).

Photon-counting lidar systems offer a promising alternative for waveform lidar technology. Individual photons received
by the instrument are time-tagged and geolocated relative to a transmitted signal (Luthcke et al., 2021). Received photons
are distinguished as signal or noise using automatic classification algorithms that are based on either histograms of detected
photons (Neumann et al., 2019) or more complex algorithms using iterative nearest-neighbor filters (Neuenschwander and
Magruder, 2019) or photon-density approaches (Herzfeld et al., 2017). Photon-counting lidars generally emit laser pulses at a
higher rate than waveform-based platforms, which results in improved along-track spatial resolution and identification of fine-
scale surface features. A drawback is that the lower transmitted energy of these systems results in more attenuation through
surfaces with low reflectance at the laser wavelength, which may limit measurement coverage. At the time of writing, ICESat-2
currently is the only active civil-space photon-counting spaceborne lidar (Markus et al., 2017).





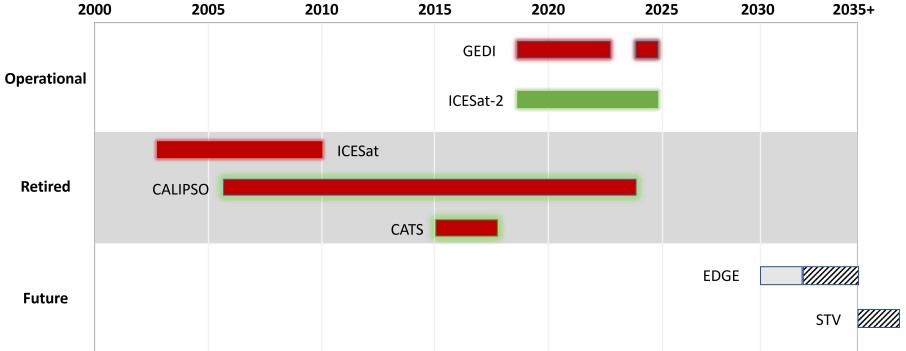

**Figure 1.** A timeline of known spaceborne lidar missions for Earth Observation from 2000-present. Bars are colored by the primary wavelength(s) for each platform. The "Operational" section includes currently active missions, whereas the "Retired" section denotes missions that are no longer active. The "Future" section indicates missions that are expected to include lidar. GEDI was placed in temporary storage aboard the International Space Station from March 2023 to April 2024. The proposed EDGE mission concept has a notional 2 year duration, but it could be extended as GEDI and ICESat-2 have been.

|  | ICESat | GEDI | ICESat-2 | DS17/GCOS |
|---|---|---|---|---|
| Sensor Type | Waveform | Waveform | Photon-counting | — |
| Wavelength | 1064 nm | 1064 nm | 532 nm | — |
| Footprint diameter | 70 m | 25 m | 11 m | — |
| Number of ground tracks | 1 | 8 | 6 | — |
| Repeat time | 2-3 times per year | 3 days | 91 days | 3-5 days |
| Max. Latitude | 86° | 51.6° | 88° | 88° (global) |
| Along-track resolution | 172 m | 60 m | 0.7 m | 100 m |
| Cross-track spacing | — | 600 m | 3.3 km | 100 m |

**Table 1.** Instrument specifications for the spaceborne lidar platforms discussed in detail in this study. The recommendations given by the 2017 Decadal Survey (DS17) and GCOS are included for comparison.

## 3 Spaceborne Lidar Missions

In this section, we describe the individual spaceborne lidar missions that have been used for snow studies. A full list of known
spaceborne lidar platforms and their operational periods may be found in Figure 1. A summary of the technical specifications
for each spaceborne lidar is given in Table 1. We recognize here that the Cloud-Aerosol Lidar and Infrared Pathfinder Satellite
Observations (CALIPSO) mission included the Cloud-Aerosol Lidar with Orthogonal Polarization (CALIOP) as part of its
scientific payload (Winker et al., 2009). The CALIOP instrument used polarized lidar backscatter to generate vertical profiles
of clouds and aerosols in the atmosphere. Similarly, the Cloud-Aerosol Transport System (CATS) was a lidar onboard the
70 International Space Station with similar science objectives to CALIPSO (McGill et al., 2015). However, both CALIPSO and



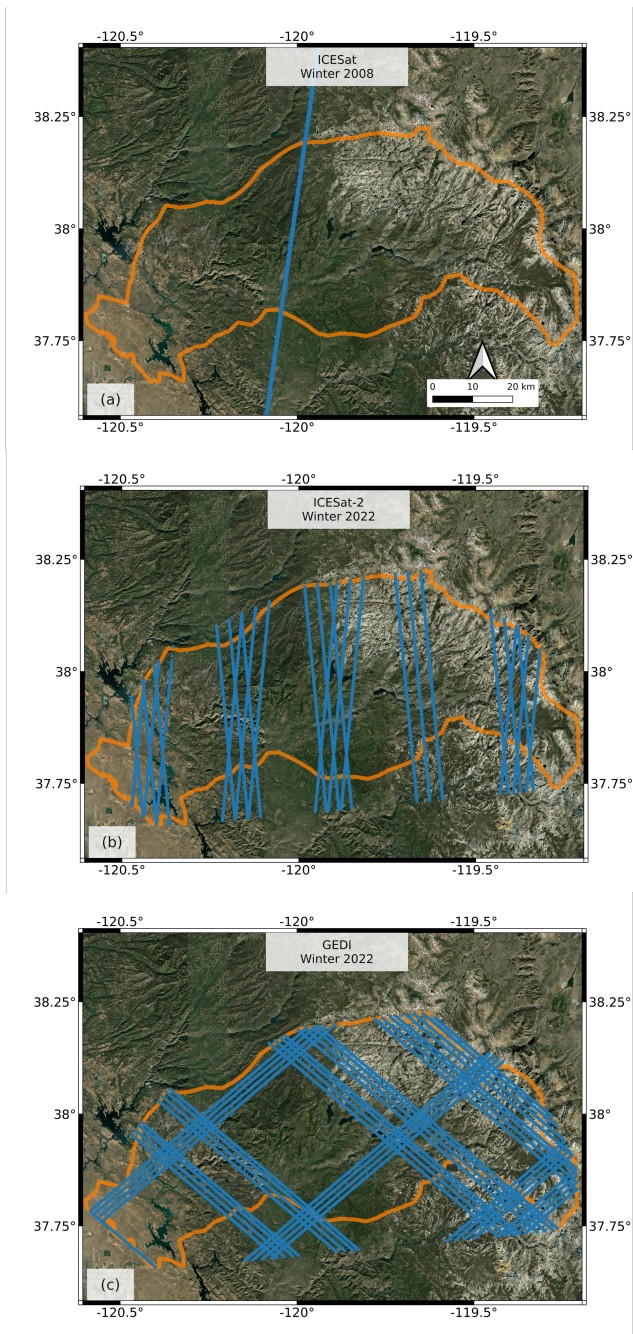

**Figure 2.** Observed satellite laser altimetry maps of the Tuolumne River Basin, CA (highlighted in orange) using Landsat imagery. The blue lines represent the total coverage of each lidar satellite for a single winter (mid-December to mid-March) season: (a) ICESat in Winter 2008, (b) ICESat-2 in Winter 2022, and (c) GEDI in Winter 2022.





CATS lacked surface elevation data products, and along-track resolution at the surface was compromised in favor of fine vertical resolution. Because of these limitations, the only snow application for CALIPSO known by the authors is the blowing snow product (Palm et al., 2017), and CATS has no known snow applications. Hence, we do not provide further discussion on CALIPSO or CATS in this paper. The Earth Dynamics Geodetic Explorer (EDGE) and the Surface Topography and Vegetation (STV) mission concepts are proposed spaceborne platforms that may include lidar as part of their respective payloads. More information about these missions may be found in Section 7.5.

### 3.1 ICESat

The original ICESat mission was launched in early 2002 with the goal of measuring interannual changes in ice elevation. Its sole onboard instrument, the Geoscience Laser Altimeter System (GLAS), primarily operated at 1064 nm, but it also included a photon-counting-based 532 nm channel to detect clouds and aerosols. The laser fired at a rate of 40 Hz with a 70 m footprint, with each measurement separated by 170 m along-track (Schutz et al., 2005). ICESat was originally conceived to operate continuously, but an engineering flaw in the three lasers required a change in the operation of GLAS to maximize laser lifetime (Abshire et al., 2005). ICESat performed a total of 18 33-day campaigns before ceasing operations in late 2009 (https://nsidc.org/sites/default/files/laseroperationalperiods.pdf).

The main altimetry products from ICESat are the GLAS/ICESat Level-2 products (GLAH12-15). Of these, the Global Land Surface Altimetry product (GLAH14) is designed for land-based elevation observations, so it is the preferred ICESat product for calculating the difference in elevation between snow-on and snow-free conditions to infer snow depth (Treichler and Kääb, 2017). However, there is approximately 70 km cross-track spacing at the mid-latitudes as a consequence of the limited observation strategy, so the coverage of ICESat is notably less comprehensive than other platforms over mid-latitude watersheds (Figure 2a).

### 3.2 GEDI

The GEDI mission was designed to measure canopy height and structure from space (Dubayah et al., 2020). GEDI was launched and added to the International Space Station (ISS) in December 2018 with a planned operational period of 2 years. The instrument operated continuously until it was temporarily placed in storage in March 2023 and returned to service in April 2024. GEDI is a full waveform lidar with a 1064 nm wavelength, similar to ICESat. The structure of the received waveform is used to distinguish between the ground and the canopy, and changes in the waveform amplitude and shape relative to the transmit pulse are used to derive canopy metrics. The GEDI footprint is 25 m in diameter, with 60 m along-track sampling from 8 beams that are spaced 600 m apart in the cross-track direction (https://gedi.umd.edu). The GEDI product relevant for snow depth is the Level-2A product, which provides along-track ground elevation and canopy height estimates. Coverage and sampling density is limited by the ISS orbit inclination of 51.6°, though dense spatial coverage is available in the mid-latitudes (Figure 2c).



## 3.3 ICESat-2

The ICESat-2 mission was launched in September 2018 to continue measurements of surface height of ice sheets and sea ice, as begun by ICESat, as well as vegetation height. Like ICESat, it carries a single instrument, the Advanced Topographic
Laser Altimeter System (ATLAS; Neumann et al., 2019). ATLAS is a photon-counting lidar that assigns a time and location (latitude and longitude) to each received photon. A single laser is split into 6 beams, with 3 beam pairs spaced by 3.3 km in the across-track direction and 90 m separation between beams within each pair. Each beam pair includes a strong beam and a weak beam to obtain sufficient coverage of high reflectivity (weak beam) and low reflectivity (strong beam) targets. The beams have an along-track sampling distance of 0.7 m, and each beam has a footprint of 11 m (Magruder et al., 2021), which allows
for significant footprint overlap. The satellite is in a polar orbit with an altitude of 500 km and a 91-day repeat cycle. The ICESat-2 orbit provides dense coverage near the poles that becomes sparser in the mid-latitudes (Figure 2b), with cross-track spacing of 2.5 km and 22 km at 80°N and 40°N, respectively. In the polar regions, data are collected along repeat ground tracks every 91-day cycle, while systematic and user-requested off-pointing at lower latitudes improve spatial coverage for vegetation mapping and for regions of interest. In the past year, the mission has pointed to prior data collections (repeat track pointing) to
enable snow applications.

ICESat-2 currently has 22 data products designed for analysis of ice sheets, vegetation, sea ice, and inland water. Of these products, three have been used in studies evaluating the potential for seasonal snow depth measurements: the Global Geolocated Photon Data product (ATL03); the Land Ice Elevation product (ATL06); and the Land, Water, and Vegetation Elevation product (ATL08). ATL03 is the base-level ICESat-2 product that is used to produce all higher-level products (Neumann et al., 2019). It
provides the highest in-track sampling at 0.7 m, but also has the least noise filtering applied. ATL06 estimates surface height by aggregating ATL03 photons into 40 m segments that overlap by 20 m (Smith et al., 2019). A windowed median is used to filter photons by quality and generate refined aggregations of surface height (Smith et al., 2018). The ATL08 product is designed to process ATL03 photons and discriminate between ground photons, noise, and several layers of tree canopies (Neuenschwander and Pitts, 2019). A median-based filtering algorithm known as the Differential, Regressive, and Gaussian Adaptive Nearest
Neighbor (DRAGANN) method is used to aggregate ground and canopy photons in 100 m segments with no overlap.

A recent development in the ICESat-2 community is SlideRule Earth, an open-source software package and an on-demand service to efficiently process ICESat-2 data in the cloud (Shean et al., 2023). In addition to facilitating standard ICESat-2 data product subsetting and delivery, SlideRule allows users to generate customizable ICESat-2 products using streamlined, parallel implementations of the ATL06 and ATL08 algorithms. Additional user controls allow for ATL03 photon filtering
based on signal confidence and the native ATL03, ATL08, and the Yet Another Photon Classifier (YAPC) photon classification approaches (Sutterley and Gibbons, 2021). It also includes support for efficient server-side sampling of large cloud-hosted DEM (e.g., ArcticDEM and REMA, 3DEP) archives, such as ArcticDEM, the Reference Elevation Model of Antarctica (REMA), and the 3-D Elevation Program (3DEP) (Porter et al., 2023; Stoker and Miller, 2022; Howat et al., 2022), as well as support for multiple GEDI products.





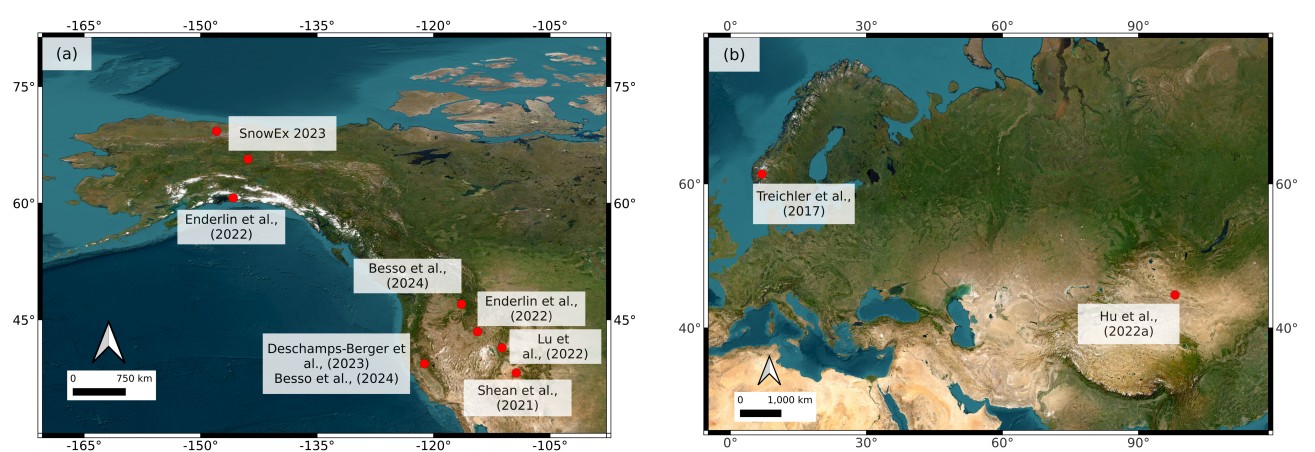

**Figure 3.** Maps of the study sites listed in Table 2, using Landsat imagery. The maps zoom in to specific regions of interest, including the western United States and Alaska (a) and Europe and Asia (b). The relevant study is given for each location. The SnowEx 2023 field sites in Alaska are also shown, as extensive ICESat-2 tasking was performed for these sites and evaluation is ongoing. Lu et al. (2022) utilized ICESat-2 granules that spanned hundreds of kilometers, so only the midpoint of these granules is shown here.

## 4 Deriving Snow Depth from Lidar Products

A list of existing studies using spaceborne lidar for snow applications is given in Table 2, with the locations or regions of interest shown in Figure 3. The listed studies perform snow depth accuracy assessments for ICESat, ICESat-2, and GEDI data products, with evaluation of land cover classification and terrain characteristics. The NASA SnowEx campaigns in 2020, 2021, and 2023 also included targeted ICESat-2 off-pointing to collect data over field sites in Colorado (2020/2021) and Alaska (2023), with the goal of an assessment of ICESat-2 snow depths in mountainous terrain, boreal forests, and tundra (Vuyovich et al., 2022). Most of the featured studies derive snow depth using differential altimetry, though other methods have been proposed by the community. When discussing the listed studies, bias refers to the difference, or residual, between spaceborne snow depths and validation depths, whereas uncertainty is a statistical range of depth values observed by a spaceborne platform. We also use the terms "accuracy" and "bias" interchangeably. We outline these approaches and findings from relevant scientific literature in the following subsections.

### 4.1 Differential Laser Altimetry

The most common method to derive snow depth from lidar is to compare two elevation datasets– one acquired when the surface was snow-free, and one acquired when the surface was snow-covered. Snow depth is assumed to be the elevation difference between the two datasets, with combined measurement uncertainty from both. This approach is known as "differential altimetry", and studies have applied this method to airborne/UAV lidar acquisitions (Deems et al., 2013; Painter et al., 2016; Harder et al., 2020; Jacobs et al., 2021) and terrestrial lidar acquisitions (Currier et al., 2019; Prokop, 2008; Revuelto et al., 2015) to




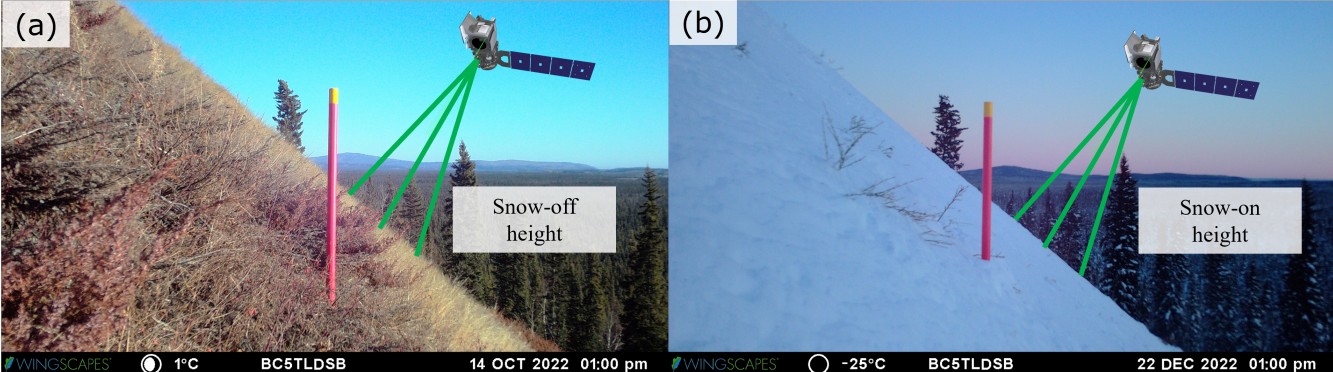

**Figure 4.** A simple example showing how snow depth is calculated using differential altimetry, with ICESat-2 used as an analogue. A snow-off elevation measurement is first obtained (a), then a snow-on measurement is taken over the same location (b). The snow depth is taken as the difference between the two height measurements. Imagery is obtained from NASA SnowEx time-lapse cameras in Bonanza Creek Experimental Forest, Alaska on 14 October, 2022 (a) and 22 December, 2022 (b).

achieve snow depth measurement accuracy of 6-17 cm, depending on the platform and study region. Figure 4 provides a visual on the measurements needed to obtain snow depth via differential altimetry. The example in Figure 4 is on sloped terrain with low-lying vegetation, which may introduce uncertainties to a depth retrieval (Section 6).

The differential method may also be used between spaceborne lidar observations and a reference snow-off DEM. The first known study to use spaceborne lidar for differential altimetry is Treichler and Kääb (2017), who used ICESat surface heights (GLAH14) and three reference DEMs to estimate snow depth in the forests of Norway. Preliminary work by Shean et al. (2021) found that GEDI snow depths had improved mid-latitude spatial coverage compared to ICESat and ICESat-2, but with larger biases and less temporal frequency. Subsequent work with spaceborne lidar has focused on ICESat-2. For instance,

Enderlin et al. (2022) used the ICESat-2 ATL06 and ATL08 products alongside airborne lidar and Worldview stereo imagery to derive snow depth over Wolverine Glacier, AK and Reynolds Creek Experimental Watershed, ID. The Tuolumne River Basin in California has been assessed by Deschamps-Berger et al. (2023) and Besso et al. (2024) using ATL06 and ATL06-SR (SlideRule) respectively, with the Airborne Snow Observatory (ASO) used as the primary DEM source. Besso et al. (2024) also examined Methow Valley, WA using SlideRule and airborne lidar from the USGS 3DEP program (Stoker and Miller, 2022).

**4.2    Other Methods**

The differential method is the most common and consistent way to derive snow depth from lidar, but Hu et al. (2022b) devised a new technique that exploits time delay due to light penetration into the snowpack (see Section 6.4) and ICESat-2 photon distributions to infer snow properties. Hu et al. (2022b) and Lu et al. (2022) deconvolved backscattered ICESat-2 photons that are reflected from the snow subsurface to derive path length distributions. These distributions are then used to estimate

snow depth, assuming that (i) terrestrial snow is a Lambertian surface and (ii) there is a sufficiently strong signal return. An



uncertainty analysis was performed by Lu et al. (2022) for both studies. When compared to a daily 4 km resolution snow depth product, the authors found snow depths with a reported accuracy of 14 cm, with 23 cm in uncertainty. Hu et al. (2022b) also used ICESat-2 backscatter to estimate snow albedo and grain size, though the accuracy of these quantities was unclear due to a lack of validation data and coarse spatial resolution. Both studies encompassed a wide range of terrain features, including land
ice, sea ice, and mountainous terrain.

It is possible to use repeat tracks of GEDI or ICESat-2 or "crossover" intersections from non-repeating tracks to perform differential altimetry measurements. Using this approach, Hu et al. (2022a) derived snow depth using intersecting ICESat-2 tracks over grasslands in Xinjiang, China, with a reported RMSE of 4 cm using ATL08. However, using cross-tracking spaceborne lidar paths consistently is difficult in the mid-latitudes due to infrequent repeat coverage, geolocation uncertainty
in repeat tracks, and possible attenuation by clouds.

Based on the existing studies, ICESat-2 has shown the most promising results for spaceborne lidar snow depth measurements (Table 3), though studies using other platforms are limited. Generally, snow depth derived from ICESat-2 have an RMSE of up to 33 cm, as determined by the studies in Table 2. ICESat is shown to perform slightly worse, with an RMSE of 47 cm reported by Treichler and Kääb (2017). GEDI has the largest bias among the three lidar platforms, with an RMSE of 101 cm over Grand
Mesa, CO (Shean et al., 2021). These platforms are also limited in their revisit frequency (ICESat) or their global coverage (GEDI). However, it must also be noted that the above assessments occur over different sites, so direct intercomparison is not possible. Because ICESat-2 has the most potential for snow depth applications, particularly over flat terrain, we will primarily focus on its measurement performance for the remainder of the paper.



| Study | Domain | Land cover | Method | Lidar product | Reference elevation | Validation |
|---|---|---|---|---|---|---|
| Treichler and Kääb (2017) | Hardangervidda, Norway | Mountain | Altimetry-DEM differencing | ICESat (GLAH14) | SRTM DEM (45 m) Kartverket DEM (10 m) | Weather stations Airborne lidar DEMs (1 m) |
| Shean et al. (2021) | Grand Mesa, CO | Forest Mountain | Altimetry-DEM differencing | ICESat-2 (ATL08, Sliderule) GEDI (L2A) | Worldview-3 (1 m) 3DEP (1 m) ASO (3 m) | Weather stations |
| Hu et al. (2022b) | Northern Xinjiang | Bare earth Grassland | Cross-track differencing | ICESat-2 (ATL08) | — | Weather stations |
| Lu et al. (2022) | Western U.S. | Forest Mountain | Backscatter deconvolution | ICESat-2 (ATL03) | — | 4 km reanalysis 24 km reanalysis |
| Enderlin et al. (2022) | Reynolds Creek, ID Wolverine Glacier, AK | Mountain glacier Forest | Altimetry-DEM differencing | ICESat-2 (ATL06, ATL08) | Airborne lidar (0.5 m, 1 m) Worldview (2 m) | — |
| Deschamps-Berger et al. (2023) | Tuolumne Basin, CA | Forest Mountain | Altimetry-DEM differencing | ICESat-2 (ATL06) | ASO (3 m) Pléiades (3 m) Copernicus (30 m) | ASO (3 m) |
| Besso et al. (2024) | Tuolumne Basin, CA Methow Valley, WA | Forest Mountain | Altimetry-DEM differencing | ICESat-2 (SlideRule) | ASO (3 m) Airborne lidar (1 m) | ASO (3 m) Weather stations |

**Table 2.** List of published studies that used spaceborne lidar for snow depth measurements, with the primary study locations, land cover types, and snow depth retrieval method given. The lidar product refers to the spaceborne lidar altimeter of interest, and the reference elevation(s) are the DEMs used for differencing.



| Study | Altimeter | Bias metric | Bias (cm) | % of Median Depth | Uncertainty metric | Uncertainty (cm) |
|---|---|---|---|---|---|---|
| Treichler and Kääb (2017) | ICESat | RMSE | 9-15 (Western forest)<br>40-64 (Eastern forest) | 5%-8%<br>34%-55% | NMAD | 103 |
| Shean et al. (2021) | GEDI<br>ICESat-2 | RMSE | 101<br>19 | — | — | — |
| Hu et al. (2022a) | ICESat-2 | RMSE | 4.2 | 27.6% | — | — |
| Lu et al. (2022) | ICESat-2 | RMSE | 14 | 28% | Standard deviation | 9.6 |
| Enderlin et al. (2022) | ICESat-2 | Residual difference | +20 (slope<0.5°)<br>-100 (slope>20°) | 29% | MAD | 60 |
| Deschamps-Berger et al. (2023) | ICESat-2 | Residual difference | -35 (ASO, slope<10°)<br>+59 (ASO, slope>40°)<br>-53 (Pléiades)<br>+53 (Copernicus) | 12%<br>20%<br>18%<br>18% | NMAD | +39 (ASO, slope<10°)<br>+148 (ASO, slope>40°)<br>+84 (Pléiades)<br>+300 (Copernicus) |
| Besso et al. (2024) | ICESat-2 | Residual difference | -5 (slope<10°)<br>-60 (slope>25°) | — | Standard deviation | 41 |

**Table 3.** Bias and uncertainties for snow depths derived from the studies in Table 2. The accuracy metrics and lidar altimeters for each study are also given. Hu et al. (2022b) is omitted from the table because accuracy and uncertainty of the method is given by Lu et al. (2022).




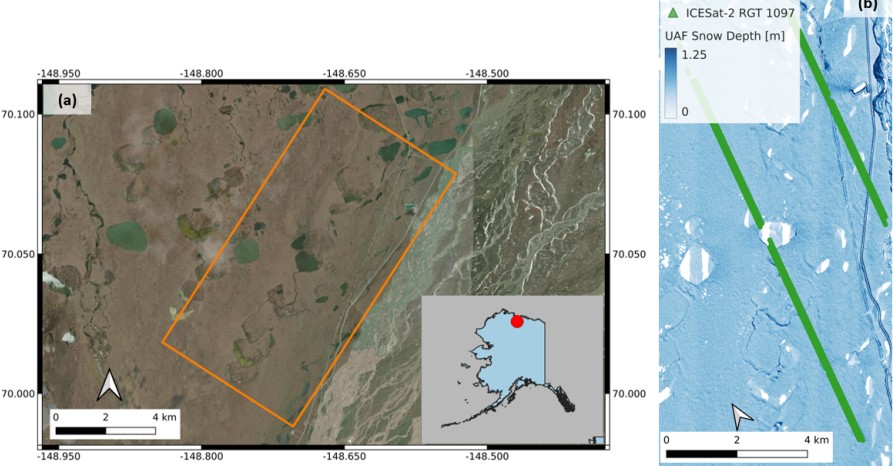

**Figure 5.** (a) Spatial domain of a SnowEx field site on the Alaskan coastal plain. The bottom right image shows the relative location of the site. (b) Snow depth product over the field site (orange box in (a)) as derived from UAF airborne lidar, with ICESat-2 RGT 1097 (March 4, 2022) given in green.

## 5 ICESat-2 Case Study in Tundra Environment

To date, studies using differential altimetry from spaceborne lidar to derive snow depth have been assessed over forest, mountains, and glaciers. Tundra environments have not been evaluated for snow depth retrievals, despite the lack of vegetation and variable terrain that complicate retrievals. Here, we perform a case study over the Arctic Coastal Plain (ACP) of Alaska to fulfill two objectives: (i) to provide an example of along-track ICESat-2 snow depths, and (ii) to illustrate the potential value of ICESat-2 snow depths over tundra environments.

The Alaskan tundra serves as a useful testbed for an accuracy assessment for multiple reasons. First, the North Slope of Alaska is at a higher latitude than previous studies, so there are a greater number of ICESat-2 tracks intersecting the region. Second, the terrain over the Alaskan tundra is very flat, so errors due to slope or DEM co-registration are assumed to be negligible, particularly if a large-area aggregated comparison is performed (see Sections 6.1 and 6.2 for a discussion on terrain and DEM errors). Finally, vegetation is limited to shrubs and tussocks. Although low-lying vegetation and permafrost melt may introduce centimeter-to-decimeter uncertainty to snow-off assessments (Section 6.3), similar snow depths should be observed between airborne lidar and spaceborne lidar, if the snow-off lidar track is flown within a year of the snow-on ICESat-2 track to account for permafrost changes.

The case study is performed using three ICESat-2 products: ATL06, ATL08, and ATL06-SR. The ATL06 and ATL08 data were accessed using icepyx, a Python tool that allows easy access of existing ICESat-2 data products (Scheick et al., 2023). ATL06-SR data was configured with parameters that would differentiate from the ATL06 and ATL08 products. The along-track resolution was set at 20 m rather than the default 40 m, and only high confidence signal photons were considered. ATL06-SR



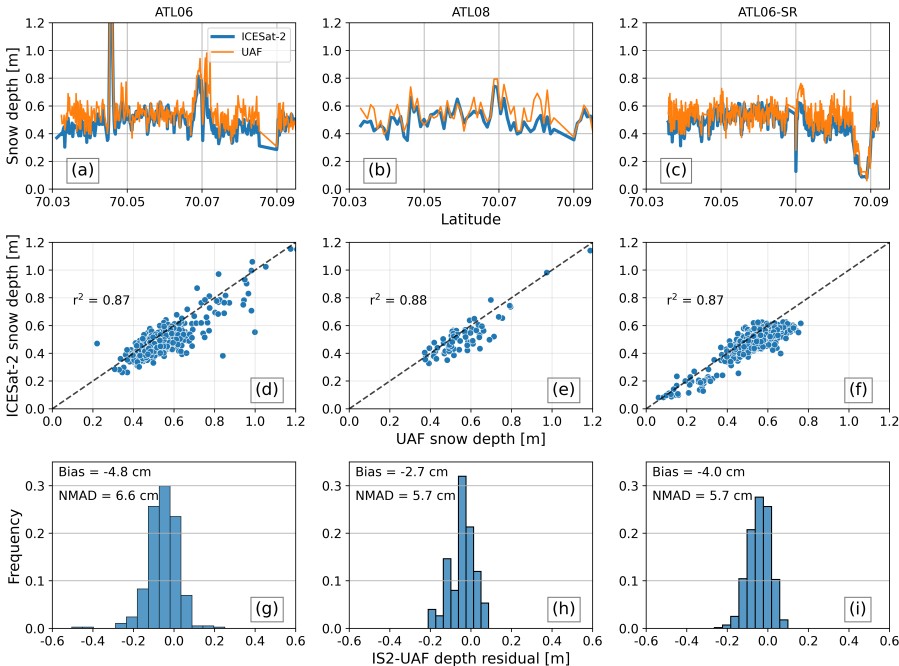

**Figure 6.** Snow depth comparisons between three ICESat-2 products (ATL06, ATL08, ATL06-SR) and UAF airborne lidar over the ACP site. The top row (a-c) shows along-track snow depth retrievals from ICESat-2 (GT2L) and UAF lidar. The middle row (d-f) is the scatter plot between the two retrieval methods. The bottom row (g-i) is the distribution of depth biases between ICESat-2 and UAF, with the median bias ("Bias") and NMAD ("spread") given for each.

also allows the use of the ATL08 classification scheme, which we used to remove photons that did not originate from the ground.

The focus of this case study, the ACP site, is located near Prudhoe Bay on the northern coast (Figure 4a), and it encompasses 210 a 75 km$^2$ tundra environment with little variation in topography. As part of the SnowEx campaign, a 1064 nm airborne lidar was flown by the University of Alaska, Fairbanks (UAF lidar) to obtain snow-on and snow-off data at 20 points per square meter and aggregated to 0.5 m resolution. Snow-off measurements were gathered on August 31, 2022 over the ACP site, and snow-on measurements were collected on March 12, 2022. An ICESat-2 overpass took place on March 4, 2022, with four of its beams overlapping the UAF lidar survey (Figure 4b). Although there is an eight-day offset between the snow-on acquisitions, snow 215 accumulation between the two dates is assumed to be negligible based on reported precipitation data at the Deadhorse Airport weather station. Temperatures were well below freezing during the observation period, so we also assume melt and sublimation were negligible. Wind speeds on and before the date of the ICESat-2 overpass were low (<10 kt), and they remained low up to the UAF lidar acquisitions.

The ICESat-2 snow depths were derived using differential altimetry between the ICESat-2 data products and snow-off data 220 from the UAF lidar. Because the UAF data is provided as a gridded raster, rather than an along-track segment like ICESat-2, we



utilized an interpolation scheme to sample UAF data along the ICESat-2 track. The UAF data were co-located to the ICESat-2 tracks using a bivariate spline interpolant, which uses available UAF snow-off and snow-on lidar data within a spatial domain to approximate surface elevation (snow-off) and snow depth (snow-on) as a function of spatial coordinates (i.e., latitude and longitude). In other words, the UAF lidar elevations and snow depths are sampled once every 20 m, 40 m, and 100 m along-track for ATL06-SR, ATL06, and ATL08, respectively, though the lidar data may not always be at the center of an ICESat-2 segment. The ICESat-2 snow depth is then taken as the difference between the ICESat-2 snow-on surface elevation ($h_{is2,on}$) and the UAF snow-off elevation ($h_{uaf,off}$):

$$d_{is2} = \Delta h = h_{is2,on} - h_{uaf,off} \tag{1}$$

Where $\Delta h$ is the ICESat-2 snow depth in meters. The co-located UAF snow depths are then used as validation for the derived ICESat-2 depths. We used the median bias and normalized median absolute deviation (NMAD) for accuracy and uncertainty metrics, and we formally define both in Section 6.1.

Figure 5 shows snow depth comparisons between ICESat-2 and UAF lidar over the coastal site. The number of snow depth data points differs between products due to the differing resolution of each: 845 depths for ATL06, 166 depths for ATL08, and 1546 depths for ATL06-SR. The different resolutions in 5a-c also illustrate variable levels of along-track variability in both ICESat-2 and UAF, with finer resolutions (ATL06 and ATL06-SR) showing greater spatial variability. However, the median snow depth is nearly the same between sampling resolutions: 48 cm for ATL06-SR and ATL06, and 50 cm for ATL08. Snow depths derived from ICESat-2 show overall agreement with UAF across all three data products, as seen in 5a-5c. The scatter plots in 5d-f suggest that ICESat-2 somewhat underestimates depth relative to the airborne lidar, particularly at larger depths. Despite the underestimations, $r^2$ values are high for all three products: 0.88 for ATL06, 0.88 for ATL08, and 0.87 for ATL06-SR . The median bias and normalized median absolute deviation (NMAD) values are at the centimeter scale (3.7-4.8 cm for bias, 5.7-6.7 cm for NMAD) for the three products, as seen in 5g-i. The biases and NMAD are comparable between products, which implies that all three products are well-suited to this environment.

Compared to previous studies for seasonal snow with different terrain and land cover types(Table 3), the biases and uncertainties observed in the tundra region are generally lower. The biases are also 10% of the total snow depth, which is within the desired accuracy defined by the Decadal Survey and GCOS (see Section 7.2). This supports the findings from previous studies, which found that spaceborne lidar snow depths improve in the absence of forests and sloped terrain (Besso et al., 2024; Deschamps-Berger et al., 2023; Enderlin et al., 2022). Lu et al. (2022) shows comparable biases to the case study over the Western United States, though more research will be needed to test their method over tundra environments relative to finer-scale measurements.





## 6 Common Error Sources

The case study in Section 5 demonstrates that accurate snow depth measurements in the tundra are possible to attain via ICESat-2. The studies in Table 2 also show that GEDI, ICESat, and ICESat-2 can retrieve snow depth over several land classes. However, snow depth accuracy and uncertainty differ between studies. The lidar platform and the retrieval method appear to have an influence, but accuracy and uncertainty also vary even between ICESat-2 studies using differential altimetry. In this section, we discuss possible sources of uncertainty for space-based lidar snow depth retrievals, including slope and terrain, vegetation, DEM source, and snowpack penetration.

### 6.1 Terrain Characteristics

Mountains have characteristic surface relief and roughness that can introduce horizontal and vertical uncertainty in lidar measurements (Deems et al., 2013). Complex topography spreads the footprint of a laser pulse non-uniformly, making precise geolocation of the received signal more difficult. Initial geolocation error is primarily related to instrument pointing errors that are exacerbated over sloped surfaces. Additional geolocation errors contribute directly to height errors as the tangent of the surface slope (Section 6.2). Pulse spreading also affects the return time of a received signal, adding uncertainty to surface elevation estimates. It is therefore critical to identify slope-based errors in both snow depth validation sources and in snow-free DEMs to quantify accuracy and uncertainty in lidar snow depth retrievals.

Several studies have quantified slope-based errors in ICESat-2 surface heights and snow depths. Wang et al. (2019) found that ICESat-2 ATL03 snow-free data had sub-meter accuracy over flat surfaces, relative to an airborne lidar over Alaska. Similar results were found in ICESat-2 ATL06 and ATL08 depths derived by Hu et al. (2022a), Enderlin et al. (2022), and Deschamps-Berger et al. (2023), with 4-20 cm in bias in all three studies at slopes < 10°. This error increases with slope and propagates to snow depths, with Enderlin et al. (2022) finding residuals and MAD values exceeding 1 m over slopes > 20°. The extent of slope-/aspect-based uncertainty is noted by Nuth and Kääb (2011), with the study noting that elevation residuals exceeded 3 m when using satellite stereo imagery over slopes > 50° and forest covers > 40%. Besso et al., (2024) demonstrated that custom ATL06 processing of ATL03 photons (SlideRule) could be used to improve ICESat-2 snow depths over mountains and dense forest, with a maximum RMSE of 33 cm and a standard deviation of 105 cm obtained. Over slopes < 10°, the authors found a median residual of 5 cm that decreased to 1 cm at slopes 0-5°. The median residuals increased to 56-60 cm at slopes > 25°, which indicates general improvement relative to previous studies.

### 6.2 DEM Accuracy and Co-Registration

The differential altimetry method to derive snow depth requires co-registration with a snow-off DEM or DTM, with different DEMs used in each of the studies highlighted above. However, DEM sources are frequently in different coordinate reference systems, and the reprojections needed prior to matching DEMs with lidar may produce geolocation uncertainties. Specifically, vertical offsets between elevation data sets are related to the magnitude of the horizontal correction and the tangent of the terrain slope angle, so geolocation offsets are generally larger over steep slopes and rugged terrain. Because the accuracy





of snow depth measurements depends on the accuracy of both the snow-off and snow-on altimetry, previous studies have calculated the most accurate spaceborne lidar snow depths using DEMs derived from airborne lidar. Deschamps-Berger et al. (2023) noted centimeter- to decimeter-scale biases over slopes below 50° even in dense forest cover when using an airborne lidar DEM, while stereographic imagery performed similarly over flat, unvegetated sites but worse over steep slopes and dense forest.

The studies in Table 2 adopt a variety of strategies to align DEMs or digital terrain models (DTMs) with ICESat, ICESat-2, or GEDI. Although DEM/DTM geolocation offsets are generally small across the studies, the varied approaches highlight the lack of a consistent method to co-register spaceborne lidar with snow-off DEMs/DTMs. The use of a DEM with broad spatial coverage, such as the 3DEP lidar or the Copernicus DEM, may enable spaceborne lidar snow studies on a regional to global scale. However, the choice of snow-free DEM/DTMs is also constrained by the need for a sufficiently high spatial resolution to resolve usable snow depths from ICESat-2. For example, Deschamps-Berger et al. (2023) found snow depth uncertainties greater than 3 m when using the Copernicus DEM, and the authors excluded it from further analysis. Besso et al. (2024) also found that the quality of the snow-off DEM was paramount, to obtain meaningful snow depth aggregates.

## 6.3 Vegetation

Tree canopy has the potential to increase snow depth errors by decreasing the strength of lidar surface returns or absorbing returns from snow underneath the canopy. Popescu et al. (2011) compared surface height measurements and canopy metrics between ICESat and airborne lidar data over the forests of eastern Texas. They found that ground height retrievals generally agreed between the two platforms, though dense vegetation may spread the returned signal pulse from ICESat and generate a height return within the tree canopy. Studies conducted by Feng et al. (2023), Neuenschwander et al. (2020), and Neuenschwander and Magruder (2019) assessed the effects of tree canopies on ICESat-2 snow-on (October - April) and snow-off (May - September) returns over boreal forests. The three studies found that the ATL08 product generally had terrain biases of -0.17 to +0.59 m over regions of dense vegetation. Interestingly, surface height retrievals had lower uncertainty over snow-covered surfaces, which was attributed to the high reflectance of signal photons of the optically-bright snow surface. We also expect that dense vegetation, such as bog understory within boreal forest environments, may be difficult for lidar signals to penetrate, thereby increasing uncertainties. However, more research will be needed over high-latitude forests to verify this claim.

Vegetative undergrowth, such as shrubs and tussocks, can introduce additional uncertainties in snow depth measurements. Results by Ilangakoon et al. (2018), Simpson et al. (2017), and Spaete et al. (2011) suggest that undergrowth can cause meter-level bias in snow-free DEMs, which can in turn produce negative snow depths in differential methods. During snow-on conditions, regions of dense undergrowth will have strong snow depth variability at small spatial scales, introducing uncertainty to lidar retrievals with comparatively large footprints (i.e., ICESat-2). For instance, results from Deschamps-Berger et al. (2023) suggest that large errors in spaceborne lidar snow depths are associated with forest cover densities over 60%. Besso et al. (2024) found increased uncertainties over Methow Valley, WA relative to the Tuolumne River Basin, CA, with denser vegetation in the valley thought to be the cause. Shrubland also proves a challenge for ground-based snow depth measurements (e.g., probing), introducing uncertainty in the validation of airborne or spaceborne lidar snow depths.





## 6.4 Lidar Penetration in Snow

Snow is weakly absorbing and highly reflective at wavelengths in the visible spectrum, resulting in a strong return signal over snow. However, a laser pulse from ICESat-2 or another 532 nm lidar may also experience scattering within a snowpack. This phenomenon, also known as "volumetric scattering", increases the time it takes for a signal to return to the detector. A modeling study conducted by Smith et al. (2018) found that volumetric scattering could bias surface elevations from 532 nm lidar by up to 50 cm when compared to 1064 nm lidar acquisitions. Observed results from Fair et al. (2024) constrain biases in ICESat-2 data to 4-7 cm at the photon level, given optical grain sizes of 1000 $\mu$m or more. However, these biases were quantified over snow and firn layers over a flat region of the Greenland Ice Sheet, and the authors speculated that it may be difficult to distinguish light penetration from other bias sources, such as topography or vegetation. Snow reflectance at NIR wavelengths is much lower than that of green wavelengths, particularly as snow ages and melts. As a consequence, volumetric scattering is not a significant issue for NIR lidar (e.g., ICESat, GEDI, most airborne lidar), though the lower reflectance reduces return signal strength (Deems et al., 2013). The case study in Section 5 examines snow prior to the melt season, so snow grain size and altimetry bias are assumed to be small.

The penetration depth may also be used to estimate snow depth, with Lu et al. (2022) giving a maximum retrievable depth of 10 m using backscatter from within the snowpack. This maximum depth was determined using snow from late winter/early spring over mountainous snow and sea ice. However, more research will be needed to assess the limits of the method, as the authors generally found depths within 1 m over their study regions.




## 7    Suggestions for Future Studies and Applications to Hydrology

### 7.1    Uncertainties in Snow Depth Retrievals and SWE Estimates

Previous snow depth studies using ICESat-2 suggest that spaceborne lidar generally works well over flat surfaces in the absence of vegetation. Sloped terrain remains a significant challenge for snow depth retrievals, so barring improvements in the absolute geolocation accuracy of spaceborne lidar, development of processing and correction algorithms is essential for spaceborne snow depths over mountainous terrain. For instance, the SlideRule project offers ways to address issues related to vegetation and mountains, including configurable segment length and spatial step size, vegetation canopy treatment, and photon filtering
(Besso et al., 2024). The choice of retrieval method may also affect accuracy. The signal convolution method by Hu et al. (2022b) and Lu et al. (2022) appears to have the best performance over the Western United States, with an RMSE of 14 cm and a standard deviation of 9.6 cm, though this is achieved by aggregating ICESat-2 observations to the resolution of a coarse reanalysis product (4 km).

Further uncertainties may be generated when converting lidar snow depths to SWE, particularly if constant snow densities
are used. Raleigh and Small (2017) show that uncertainty in snow density estimates is the largest source of error in SWE estimates when converting lidar snow depths greater than 60 cm. Snow depth from these sources could be combined with ground-penetrating radar (GPR), physically-based and semi-empirical models, or in-situ snow densities to better estimate SWE (McGrath et al., 2022; Meehan et al., 2023; Webb et al., 2018).

### 7.2    Reported Accuracy Metrics

The studies outlined here generally use the mean or median difference to quantify biases in snow depths. However, the metrics used to quantify bias (accuracy) and uncertainty differ between studies (Table 3). Lu et al. (2022), Treichler and Kääb (2017), and Besso et al. (2024) also used the root mean square error (RMSE) to estimate snow depth errors relative to validation measurements. Uncertainty metrics are more varied across studies, including the standard deviation (Lu et al., 2022; Besso et al., 2024), interquartile range (Enderlin et al., 2022; Treichler and Kääb, 2017), the median absolute deviation (Enderlin
et al., 2022), and the normalized median absolute deviation (Deschamps-Berger et al., 2023). Each uncertainty metric assesses snow depth variability differently, so it is difficult to compare results between studies unless random error with a normal distribution is assumed.

Although the snow depth residuals in Figure 5 have a near-normal distribution, this is uncommon in other ICESat-2 snow depth studies, so robust statistical measures are needed. In our case study (Section 4.3), we selected the median residual and
normalized median absolute deviation (NMAD) to assess snow depth accuracy and uncertainty. These metrics can be computed using the following equations:

$$\delta d_i = d_{is2,i} - d_{uaf,i} \tag{2}$$



$$m_{\delta d} = median(\delta d) \tag{3}$$

$$NMAD = 1.4826 * median(|\delta d_i - m_{\delta d}|) \tag{4}$$

Where $\delta d$ is the snow depth residual at point $i$, $m_{\delta d}$ is the median snow depth residual over all points, $d_{is2}$ is the ICESat-2 snow depth, and $d_{uaf}$ is the validation snow depth (UAF lidar in Section 4.3).

These metrics were used to minimize the influence of outliers in the data, which are otherwise common in fine-scale datasets such as ICESat-2 and the UAF lidar. The NMAD also has the advantage of being equivalent to the standard deviation if the underlying data has a normal distribution, provided a sufficiently large number of observations (Höhle and Höhle, 2009). Due

to the frequency of outliers, we recommend using the median bias and NMAD to quantify along-track spaceborne lidar snow depth error metrics in future studies. Estimating the percent error compared to total snow depth will also be useful to compare against 2017 Decadal Survey requirements.

### 7.3    Feasibility of Spaceborne Lidar to Support Snow Hydrology Science and Applications

The snow observation requirements, as reported by the 2017 Decadal Survey and the Global Observing System Essential

Climate Variables (GCOS ECVs), advocate for repeat global SWE measurements every 1-5 days with 10-20% accuracy (Table 4). Our literature review and case study demonstrate that ICESat-2 can provide high-resolution snow depths with centimeter-level accuracy under ideal conditions. Despite the shortcomings discussed in Section 6, progress has been made on improving snow depth accuracy from spaceborne lidar. Kwon et al. (2021) conducted an observing system simulation experiment (OSSE) to determine the assimilated snow depth accuracy needed to improve snow models. It was found that an error threshold of 40

380    cm was needed to provide beneficial improvements to modeled SWE. This level of accuracy cannot be achieved through the current methods using ICESat and GEDI (Table 3). However, ICESat-2 is shown to perform within 40 cm of error, given (i) the local slope is less than 20° and (ii) an accurate, high-resolution snow-off DTM is used (Deschamps-Berger et al., 2023). Besso et al. (2024) also found that filtering ICESat-2 noise photons using SlideRule improved accuracy over complex terrain.

Spaceborne lidar is currently unable to fulfill the revisit times necessary to achieve global SWE observations every 1-5 days.

Snow evolves throughout the season with accumulation events approximately every 5-7 days, or in strong episodic events (Pomeroy et al., 1998). Snow melt events occur over a period of days to months depending on the landscape and snow depth (Liston, 2004; Musselman et al., 2017). Capturing the timing of snow melt is especially critical to inform streamflow forecasting and water management (Anghileri et al., 2016; Gagliano et al., 2023). Spatial coverage of snow observations is also important for capturing the spatial variability of the snowpack. Currently, ICESat-2 direct repeats are every 91 days, though basin-scale

repeats have shorter revisit times. GEDI has more frequent repeats (~3 days), but only over specific tracks. Kwon et al. (2021) showed that, with this limited coverage, there was minimal benefit when assimilating spaceborne lidar even with a hypothetical wide swath platform, though methods to extrapolate information from the lidar swath to a wider domain were not used.



| Snow variable | DS Requirement | GCOS Requirement | Does spaceborne lidar fulfill objective? | Comments |
|---|---|---|---|---|
| Snow depth | — | 5 km resolution<br>1 day revisit time<br>25 mm uncertainty | Yes<br>No<br>Yes | Accuracy is possible over flat terrain. Other environments have decimeter accuracy. Revisit time is not achievable. |
| Snow cover | 1-10 km resolution<br>1-2 times per day | 500 m resolution<br>1-4 times per day | Needs research | Snow cover metrics have been proposed, but not developed. Optical imagery is preferred. |
| SWE | 4 km resolution<br>100 m resolution (mountains)<br>3-5 day revisit time<br>10-20% accuracy | 5 km resolution<br>—<br>1 day revisit time<br>30% accuracy (mountains) | Yes<br>Yes<br>No<br>Yes | Gives snow depth, needs density observations or models to derive SWE. Resolution is along-track; across-track is coarser. |
| Optical properties | 30 m resolution | N/A | Needs research | Optical property retrievals have been proposed but not developed. |

**Table 4.** A summary of recommended specifications for four snow variables, and the feasibility of spaceborne lidar to fulfill these requirements from the 2017 Decadal Survey (DS) and Global Climate Observing System (GCOS). Requirements that have the potential to be fulfilled, but do not have published literature relevant to spaceborne lidar, are marked as "Needs research". Caveats for each snow variable are given in the "Comments" column. The GCOS requirements (*) are in the process of being updated, so the values here are subject to change.

A current limitation in achieving global snow depth observations from spaceborne lidar is the need for an accurate snow-off DEM when using the differencing approach, ideally from spaceborne lidar. Deschamps-Berger et al. (2023) showed that less 395 than 2.5% of the Tuolumne Basin was covered by ICESat-2 during the snow-off season across three years. Additionally, no currently available global DEM product has demonstrated the ability to achieve accurate snow depths when combined with spaceborne snow-on lidar observations. Until a global DEM product with sufficient accuracy and resolution is available, the utility of spaceborne lidar for mid-latitude snow depth observations will be limited to locations with a high-quality snow-off DEM available.

An important consideration for future hydrologic applications is data latency. Current spaceborne lidar missions have a data latency on the order of months (a minimum of 1.5 months for ICESat-2, 4 months for GEDI), which hinders their utility for





operational snow monitoring. To meet data needs for sea ice and vegetation applications, ICESat-2 provides expedited "quick look" data sets for several of its products. These quick look products are released three days after acquisition and downlink, though they do not include the pre-processing used to correct ICESat-2 orbital positioning and pointing. Otherwise, an ideal
spaceborne lidar mission would include a low data latency with pre-processing applied, especially if regular monitoring of a watershed is desired.

### 7.4 Combining Spaceborne Lidar Data and Hydrologic Models

Some of the limitations in snow depth retrievals from spaceborne lidar may be overcome with hydrologic models, in particular the limited coverage and repeat times. Previous studies have demonstrated that assimilation of airborne lidar observations
can improve modeled estimates of snow depth, density, and SWE (Hedrick et al., 2018; Margulis et al., 2019; Smyth et al., 2019). These studies show that the greatest model improvement comes from one high-quality map of snow depth near the peak snowpack, suggesting that within a model framework, temporally continuous satellite data may not be necessary. However, lidar platforms with large temporal gaps are unlikely to capture critical snow evolution periods, such as the time of peak snow. Due to the low spatial coverage of spaceborne lidar overpasses, snow depth derived from satellite altimetry will likely be
most useful for modeling if the limited extents of snow depth observations are used to infer snow depth in adjacent pixels to correct models. Multiple approaches for this application exist, including multidimensional Kalman filters/smoothers (Alonso-González et al., 2023; Magnusson et al., 2014); statistical approaches like kriging (Collados-Lara et al., 2017); interannual snow depth, snow cover, and SWE persistence patterns (Pflug et al., 2022); and other machine learning approaches (Cui et al., 2023; Guidicelli et al., 2024; Liu et al., 2024). For any of these methods, snow depth observations over multiple elevation
regimes, aspects, and land cover types would contain more information than repeat airborne lidar observations over a single region (Margulis et al., 2019), as they would capture the widest variability in snow depth, snow density, and snowpack state. Because precipitation biases are responsible for significant errors in snow models (Henn et al., 2018; Pflug et al., 2021; Smyth et al., 2020; Wayand et al., 2015), accurate lidar observations during peak SWE and prior to melt onset would be useful to correct over- or under-estimation of snow accumulation. For instance, Guidicelli et al. (2024) found that assimilation of snow
depth from a single ICESat-2 track from the late accumulation season improved estimated peak snow amounts. Assimilation of spaceborne lidar snow depths would also be beneficial during the melting season, where radar-based retrievals are less effective.

  Provided a snow-off DEM, an ICESat-2 track can theoretically be used alongside historic SWE data to determine SWE over a large watershed of length scale 1-10 km. Assuming that SWE values are spatially correlated (i.e. all SWE values are above
430 or below spatial or temporal averages), the broader watershed domain can be updated with a single ICESat-2 track. Accurately transforming ICESat-2 snow depth measurements to usable SWE estimates will require snow density observations, quantification of measurment uncertainty, and correlations between location-specific depth and domain-wide depth. Additionally, ensemble-based data assimilation frameworks, such as those described above, are ideal to accurately assimilate ICESat-2 depths into models. Besso et al. (2024) demonstrate that the median snow depth has little bias in the Tuolumne Basin, so even infre-
435 quent ICESat-2 snow depths could be used to accurately infer SWE throughout the snow season (Margulis et al., 2019). These





findings were supported by Mazzolini et al. (2024), who performed a data assimilation study to improve reanalysis-derived SWE measurements using ICESat-2 snow depths. The ICESat-2 community has made data processing tools and workflows readily available through multiple hackweeks (Arendt et al., 2020), so the modeling community can easily conduct further data assimilation studies using ICESat-2 data.

Current SWE reconstruction methods use a combination of hydrologic models and reanalyses such as the ECMWF Reanalysis v5 (ERA5) and Modern-Era Retrospective analysis for Research and Applications, v2 (MERRA-2), but this approach can only be used after the water year has occurred. Additionally, hydrologic models are improving constantly, but biases due to both modeling and forcing errors have significant implications on estimates of snow water resources (Kim et al., 2021; Mudryk et al., 2023; Raleigh et al., 2015). In many regions with significant snowfall, these modeling errors are chiefly caused by pre-

cipitation biases (Henn et al., 2018; Hughes et al., 2020; Lundquist et al., 2015; Pflug et al., 2021). As a consequence, models experience divergence in simulated snow accumulation, in heat content, and in the timing of seasonal snowmelt onset and snow disappearance. However, previous studies have also shown that there is often repeatability in snow patterns on an interannual scale (Deems et al., 2008; Pflug et al., 2022, 2021; Pflug and Lundquist, 2020; Schirmer et al., 2011; Schirmer and Lehning, 2011; Sturm and Wagner, 2010), so consistent observations near times of peak SWE will ideally bias correct modeled snow

estimates at larger spatial scales.

### 7.5   Future Satellite Laser Altimetry Missions

The discussion in this paper focuses on currently operational satellite missions, primarily the ICESat-2 mission. However, there are future spaceborne lidar altimetry missions that may provide additional opportunities for snow depth retrievals upon launch. The first such lidar mission is the proposed EDGE mission, which is a NASA Earth System Explorer concept that was

selected for Phase A study in May 2024. EDGE proposes a swath-mapping lidar with <3m horizontal geolocation accuracy for low slopes (https://edge.ucsd.edu/instrument/). EDGE will be a major technological advancement over currently operational satellite altimetry missions, with 40 beams distributed across five 8-beam mini-swaths that offer dense sampling in both the along-track and across-track directions. While the EDGE concept has been optimized for terrestrial ecosystem structure and ice elevation measurements, EDGE will also offer precise seasonal snow depth measurements using the same methods outlined in earlier sections. EDGE will also offer improved canopy penetration compared to ICESat-2, and will capture the spatial

variability of snow depth across multiple relevant spatial length scales. If selected for continued development, EDGE is slated to launch in ~2030.

A second mission concept with a proposed lidar payload is the Surface Topography and Vegetation (STV) mission, which was conceived as a set of priority targeted observables for incubation study by the 2017 Earth Science Decadal Survey. The

initial STV Study Team report (Donnellan et al., 2021) identified seasonal snow depth as one of 5 priority observables. Candidate measurement strategies include some combination of lidar, radar, and stereo photogrammetry, with candidate architecture including both satellites and airborne platforms. Multiple next-generation satellite lidar concepts, such as the Concurrent Artificially-intelligent Spectrometry and Adaptive Lidar System (CASALS), are under consideration, with ongoing technology



maturation efforts underway in advance of the upcoming 2027 Decadal Survey. A launch for an STV observable is targeted for
the mid-2030s.





## 8   Conclusions

With recent trends in climate change, it is becoming increasingly important to monitor available freshwater sources. Snow is a vital freshwater source for billions of people across the globe, so methods to monitor snow water equivalent and snow depth are needed. In-situ and airborne instruments provide high-quality measurements of snow depth and SWE at select watersheds, but spaceborne methods will be required to obtain routine observations at larger spatial scales. Spaceborne lidar also has the potential to play a role in an overall global snow observing strategy by providing high-resolution snow depth observations, particularly during the season when other snow remote sensing techniques struggle. Recent developments show that spaceborne lidar provide useful snow depth data in areas where the local slope is below 20° and bare earth DEMs/DTMs are available. Models which can assimilation observations and fill gaps in space and time are critical to utilizing spaceborne lidar for hydrological applications, though the exact measurement requirements to add value still need to be determined. There are currently two spaceborne lidar technologies available for snow applications: GEDI and ICESat-2. Of the two platforms, ICESat-2 generally offers better accuracy, greater coverage of high-latitude sites, and more continuous spatial coverage. However, Besso et al. (2024) demonstrated that customized processing of ICESat-2 products using SlideRule will be important to minimize uncertainties across variable terrain and land cover types. We recommend using median depth and the normalized median absolute deviation (NMAD) when assessing snow depth accuracy and uncertainty to reduce the influence of outliers.

There remain a few science questions that we leave for future studies. First, global snow depth observations from spaceborne lidar will not be possible until an accurate, high-resolution, DEM over regions with seasonal snow is available. To improve accuracy, a greater understanding of the geolocation accuracy of reference DEMs, and how said accuracy changes over time, is needed. This limitation highlights the need for an open-access, high-resolution global DEM, as current DEMs are limited in total coverage (ArcticDEM) or in spatial resolution (Copernicus DEM). A greater understanding of acceptable spatial resolution for reference DEMs is also needed to capture spatial variability in snow depth. At a regional scale, snow-free acquisitions are infrequent, and there is a risk of significant landscape changes occurring between DEM acquisition and spaceborne lidar retrieval, particularly in areas with melting permafrost. Second, more research is needed to validate spaceborne lidar snow depths against in-situ and airborne methods. Airborne methods such as the ASO campaigns will provide valuable, high-resolution snow depths for assessment and monitoring of mid-latitude watersheds. In-situ validation will be especially important to characterize uncertainties due to vegetation, which may be difficult to quantify with airborne and other space-based methods. Finally, more research is needed to determine how much of a watershed or basin must be sampled to improve modeled estimates. Combining spaceborne lidar observations with physical and statistical models may help fill observational gaps in an overall global snow observing strategy.



*Code and data availability.* The code used for the case study in Section 5 may be found at the following Zenodo link: https://doi.org/10.5281/zenodo.13852000. The UAF lidar data (Larsen, 2024) and the ATL06/ATL08 data products (Smith et al., 2019; Neuenschwander and Pitts, 2019) were obtained from the National Snow and Ice Data Center (NSIDC). Documentation and instructions on using may be found in (Shean et al., 2023). The camera imagery in Figure 4 was obtained during the SnowEx 2023 campaigns, and is currently pending upload to NSIDC.

*Author contributions.* ZF and CV were responsible for the conception and drafting of the manuscript. TN and DS provided insight on the lidar platforms discussed in the manuscript, and gave feedback on the manuscript. JP, HB, and JL gave perspectives on the modeling and data assimilation aspects of the paper. EME, KZ, HB, CDB, and DT contributed to Section 4 of the manuscript, and provided valuable insight for Sections 6 and 7.

*Competing interests.* At least one of the (co-)authors is a member of the editorial board of The Cryosphere.

*Acknowledgements.* This review paper was supported by NASA Postdoctoral Program grant 168273S and Co-operative Research grant 012454. Additional support for co-authors Hannah Besso and Jessica Lundquist was given by NASA grant 80NSSC20K1293.



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
