# Peer review of "Review article: Using spaceborne lidar for snow depth retrievals: Recent findings and utility for global hydrologic applications"

_EGUsphere, 2024_

## Author Response (AR1)

Authors' Response to Reviewer Comments for "Review article: Using spaceborne lidar for snow depth retrievals: Recent findings and utility for global hydrologic applications"

The authors thank both reviewers for their insightful comments for the improvement of this manuscript. For this response, the original comments are in orange, and the authors' responses are in blue. We also acknowledge the comments from a community member, which we also address here.

**Reviewer #1**

The literature review is broad and offers a valuable overview of available approaches, and the explanation of basic principles serves as introduction for readers new to the technique. However, the level of detail and technical terminology is at times quite dense. I recommend adding a detailed schematic figure, similar to Figure 4, that visually represents key concepts such as along-track resolution, across-track resolution, and beam footprint for example. This would aid in comprehension and serve as a quick reference.

This is a good suggestion. We added a new figure early in the text to coincide with the basic lidar principles. This figure outlines the three concepts raised by the reviewer (along-track resolution, across-track resolution, beam footprint).

Figure XX. (a) Sample lidar swath (orange) to demonstrate along-track resolution, across-track resolution, and footprint size. In this example, the swath width (across-track resolution) is approximately 120 m, the footprint size is 10 m, and the along-track resolution is 50 m. These values do not reflect any active or proposed spaceborne lidar mission, and were arbitrarily selected for visualization purposes.

The discussion regarding error sources is informative. An easily accessible recap table summarizing these uncertainty sources and their associated uncertainties would enhance the reader's ability to grasp and compare the contributions of each factor.

This is another good suggestion. We added a table that summarizes each error source: terrain characteristics, DEM accuracy, vegetation, and lidar penetration. A range of potential biases is also given for each error source. Note that the final table looks more refined when compiled into Latex; the below table is for easy reference for the reviewers.

Table 1. A summary of the error sources discussed in Section 6. The given biases in the right column represent maximum biases reported in available literature, though because undergrowth (\*) has not been formally assessed for DEM generation, the given value is speculative.

| Error Source                         | Impact to Lidar                                                                                                    | Expected Biases (cm)                                   |
|--------------------------------------|--------------------------------------------------------------------------------------------------------------------|--------------------------------------------------------|
| Terrain characteristics              | Complex topography (surface roughness, slope) makes precise geolocation of the return signal difficult.            | >100 (ATL06/ATL08)
60 (ATL06-SR)
for slope > 20° |
| DEM accuracy and co-
registration | Reprojecting to match reference DEMs can cause geolocation uncertainties.                                          | <10 (lidar DEMs)
>100 (coarse DEMs)                 |
| Vegetation                           | Dense vegetation canopies weaken the return signal. Undergrowth introduces uncertainties in snow-free DEMs.        | ~60 (forest cover ~60%)
~100 (heavy undergrowth)*   |
| Lidar penetration in snow            | The lidar signal experiences scattering within a snowpack, increasing the time it takes to return to the detector. |

Line 371. I like the idea of relative error, which will help contextualize the order of magnitude of the actual bias of the methodology. Could other metrics commonly used by the hydrological community, for instance Kling Gupta Efficiency (KGE) use in this context?

To the authors' knowledge, the KGE metric is not commonly used in the snow science community. We acknowledge its potential value for observation-model intercomparisons, though because the studies in Table 2 do not use KGE or other metrics, we will leave further discussion on their usage for a future study.

**Line 410. Could you include some global hydrological studies? If they exist.**

Most snow studies tend to focus on the watershed scale, meaning that hydrologic impacts are also examined at the watershed scale. So, global assessments of snow are limited beyond model intercomparisons and reanalysis products. We added citations to ECMWF's global snow analysis paper, the GlobSnow model, and the Earth System Model-Snow Model Intercomparison Project (ESM-SnowMIP):

"Some of the limitations in snow depth retrievals from spaceborne lidar may be overcome with hydrologic models and reanalysis products, in particular the coverage and repeat times. Initiatives such as the Earth System Model-Snow Model Intercomparison Project (ESM-SnowMIP), the European Center for Medium-Range Weather Forecasts (ECMWF) operational snow analysis, and the GlobSnow model have performed assessments of snow observations and model outputs over the Northern Hemisphere (Drusch et al., 2004; Krinner et al., 2018; Luojus et al., 2021). In addition, previous studies have demonstrated..."

Line 479. The authors note that spaceborne lidar provides reliable snow depth data in areas where the local slope is less than 20°. Would it be possible to give a percentage of the global seasonal snow area where this requirement is fulfilled?

This is a great question. It would be difficult to give a percentage on a global scale, but we note in the conclusion that this would lead to good representation in flat regions (e.g., tundra) and less coverage in mountainous regions.

"Recent developments show that spaceborne lidar provides useful snow depth data in areas where the local slope is below 20° and bare earth DEMs/DTMs are available. Over regions with consistent winter snow cover, these constraints are consistent with the Arctic tundra or valleys or plateaus in mountainous regions."

In general, the location of the Figures does not help the reader to follow the text. I suggest placing them after they are referred to in the text and close to the citation.

We rearranged placement of the figures accordingly.

Figure 1. I recommend including a legend or further explanation about the colours used. Now, it is not clear what the authors mean by primary wavelength.

We added more information in the caption to provide context on the colors.

"...Bars are colored by the primary wavelength(s) for each platform: red for 1064 nm and green for 532 nm."

Figure 2. The caption is not clear. What do you refer to by observed satellite laser altimetry maps using Landsat? Is Landsat used here for altimetry?

We revised the caption to prevent confusion.

"Observed satellite laser altimetry maps of the Tuolumne River Basin, CA (highlighted in orange), with Landsat imagery mosaics used as a basemap."

Figure 3. It is not clear to me what the term "using Landsat imagery" means. I assume you want to indicate that the basemap used was taken from Landsat. If it is the case, I think it is not needed; if not, please clarify.

This is correct. We clarified this in the updated caption.

"Maps of the study sites listed in Table 2, using a basemap derived from Landsat imagery."

Line 214. Figure 4b is not correctly cross-referenced.

Fixed.

**Community Comments**

This manuscript is presenting a review on using spaceborne lidar for snow depth retrievals. The authors gathered the recent studies on retrieving snow depth from spaceborne lidar data. They presented a case study over the tundra of Alaska to present accuracy estimates for several current methods. The manuscript is written well and presents the current status of research on using spaceborne lidar in retrieving snow depth to be used in operational hydrological studies. I recommend acceptance after minor revisions. There are some minor comments listed below:

We thank the community member for their comments on our manuscript. Because many of the comments given here were addressed in the reviewer responses, we summarize our changes to the community member's suggestions here.

- Line 209, Figure 4a must be Figure 5a
- Line 232, Figure 5 must be Figure 6
- Line 358, Figure 5 must be Figure 6
- In Figure 3 Hu et al. (2022a) must be Hu et al. (2022b) which is also given in Table 2.

**These typos were all fixed.**

• It would be good to include the size of the study areas in km2 in Table 2, it may give an idea in using these observations for hydrological modelling that is stated in the Conclusion part.

Thank you for the suggestion. We added the areal coverage for all but the Hu et al., (2022b) and Lu et al., (2022) paper, because they focused on ICESat-2 tracks rather than specific domains.

- In Figure 6, it would be good to present the common points from ATL06, ATL08 and ATL06-SR with a different colour to present the consistency of the products in retrieving the snow depth.
- There is so much spatial variation in ATL06 and ATL06-SR products. Are these noises or that spatial variation exists along the track. Especially the snow depth larger than 1.2 m between 70.03° and 70.05° in Figure ATL06 is questionable. What is the reason to have this large snow depth. In scatterplot of ATL08 1.2 m is seen but it is not seen in snow depth figure of ATL08. 1.2 m snow depth is not presented in ATL06-SR snow depth and scatterplot figures.
- What is the reason to have a constant snow depth around 70.09 degree in ATL06 snow depth figure. It seems there is a gap of UAF snow depths in this area and an interpolation is applied in this region.
- It would be good to include ground elevation in snow depths of Figure 6. It would give us an idea how the ground elevation is changing along the track.

In response to other reviewers, Figure 6 (now Figure 7) has been overhauled to show an example elevation/snow depth retrieval using ICESat-2, specifically with the ATL06-SR product. The new example has filtering applied to remove faulty values, such as the 1.2 m noted by the community member. The changes also address the remaining comments on Figure 6 by (i) removing ATL06 from the figure, and (ii) adding surface elevation as a subplot to the figure.

• "Spaceborne lidar is currently unable to fulfill the revisit times necessary to achieve global SWE observations every 1-5 days." I think this sentence is not correct. We can retrieve snow depth from spaceborne lidar but not snow density. Even data availability can be every 1-5 days, how can the snow depth retrieved from lidar can be used to obtain SWE?

This is correct – spaceborne lidar provides snow depths, but not snow density. So, it technically provides a SWE precursor rather than SWE. We revised this sentence to provide better clarification:

"Spaceborne lidar is currently unable to fulfill the revisit times necessary to retrieve snow depths for global SWE observations every 1-5 days."

**New References**

Dozier, J., (1989). Spectral signature of alpine snow cover from the landsat thematic mapper. Remote Sensing of Environment, 28, 9-22, https://doi.org/10.1016/0034-4257(89)90101-6

Drusch, M., D. Vasiljevic, and P. Viterbo, 2004: ECMWF's Global Snow Analysis: Assessment and Revision Based on Satellite Observations. *J. Appl. Meteor. Climatol.*, **43**, 1282–1294, https://doi.org/10.1175/1520-0450(2004)043<1282:EGSAAA>2.0.CO;2.

Elder, K., Rosenthal, W., Davis, R. E., (1998). Estimating the spatial distribution of snow water equivalence in a montane watershed. Hydrol. Process., 12, 1793-1808. https://doi.org/10.1002/(SICI)1099-1085(199808/09)12:10/11%3C1793::AID-HYP695%3E3.3.CO;2-B

Gascoin S., Grizonnet, M., Bouchet, M., Salgues, G., Hagolle, O., (2019). Theia Snow collection: high-resolution operational snow cover maps from Sentinel-2 and Landsat-8 data. Earth Syst. Sci. Data, 11, 493–514, https://doi.org/10.5194/essd-11-493-2019.

Hall, D. K., Riggs, G., A., Salomonson, V. V., DiGirolamo, N. E., Bayr, K. J., (2002). MODIS snow-cover products. Remote Sensing of Environment, 83, 181-194, <a href="https://doi.org/10.1016/S0034-4257(02)00095-0">https://doi.org/10.1016/S0034-4257(02)00095-0</a>.

Kinar, N. J. and Pomeroy, J. W., (2015). Measurement of the physical properties of the snowpack. Reviews of Geophysics, 53, 481-544. https://doi.org/10.1002/2015RG000481

Kokhanovsky, A., Lamare, M., Danne, O., Brockmann, C., Dumont, M., Picard, G., Arnaud, L., Favier, V., Jourdain, B., Le Meur, E., Di Mauro, B., Aoki, T., Niwano, M., Rozanov, V., Korkin, S., Kipfstuhl, S., Freitag, J., Hoerhold, M., Zuhr, A., ... Box, J. E. (2019). Retrieval of Snow Properties from the Sentinel-3 Ocean and Land Colour Instrument. *Remote Sensing*, *11*(19), 2280. https://doi.org/10.3390/rs11192280

Krinner, G., Derksen, C., Essery, R., Flanner, M., Hagemann, S., Clark, M., Hall, A., Rott, H., Brutel-Vuilmet, C., Kim, H., Ménard, C. B., Mudryk, L., Thackeray, C., Wang, L., Arduini, G., Balsamo, G., Bartlett, P., Boike, J., Boone, A., Chéruy, F., Colin, J., Cuntz, M., Dai, Y., Decharme, B., Derry, J., Ducharne, A., Dutra, E., Fang, X., Fierz, C., Ghattas, J., Gusev, Y., Haverd, V., Kontu, A., Lafaysse, M., Law, R., Lawrence, D., Li, W., Marke, T., Marks, D., Ménégoz, M., Nasonova, O., Nitta, T., Niwano, M., Pomeroy, J., Raleigh, M. S., Schaedler, G., Semenov, V., Smirnova, T. G., Stacke, T., Strasser, U., Svenson, S., Turkov, D., Wang, T., Wever, N., Yuan, H.,

Zhou, W., and Zhu, D.: ESM-SnowMIP: assessing snow models and quantifying snow-related climate feedbacks, Geosci. Model Dev., 11, 5027–5049, https://doi.org/10.5194/gmd-11-5027-2018, 2018.

Luojus, K., Pulliainen, J., Takala, M. *et al.* GlobSnow v3.0 Northern Hemisphere snow water equivalent dataset. *Sci Data* **8**, 163 (2021). https://doi.org/10.1038/s41597-021-00939-2

Meyer, R., Schødt, M. P., Rasmussen, M. L., Andersen, J. K., Dømgaard, M., and Bjørk, A. A., (2025). A new method for large scale snow depth estimates using Sentinel-1 and ICESat-2, EGUsphere [preprint], https://doi.org/10.5194/egusphere-2024-3850.

Oveisgharan, S., Zinke, R., Hoppinen, Z., and Marshall, H. P., (2024). Snow water equivalent retrieval over Idaho – Part 1: Using Sentinel-1 repeat-pass interferometry, The Cryosphere, 18, 559–574, https://doi.org/10.5194/tc-18-559-2024.

Painter, T. H., Rittger, K., McKenzie, C., Slaughter, P., Davis, R. E., Dozier, J., (2009). Retrieval of subpixel snow covered area, grain size, and albedo from MODIS. Remote Sensing of Environment, 113, 868-879, https://doi.org/10.1016/j.rse.2009.01.001.

Proksch, M., Rutter, N., Fierz, C., Schneebeli, M., (2016). Intercomparison of snow density measurements: bias, precision, and vertical resolution. The Cryosphere, 10, 371-384. https://doi.org/10.5194/tc-10-371-2016

Riggs, G. A., Hall, D. K., Román, M. O., (2017). Overview of NASA's MODIS and Visible Infrared Imaging Radiometer Suite (VIIRS) snow-cover Earth System Data Records, Earth Syst. Sci. Data, 9, 765–777, https://doi.org/10.5194/essd-9-765-2017.

Tedesco, M. & Jeyaratnam, J. (2019). *AMSR-E/AMSR2 Unified L3 Global Daily 25 km EASE-Grid Snow Water Equivalent*. (AU\_DySno, Version 1). Boulder, Colorado USA. NASA National Snow and Ice Data Center Distributed Active Archive Center. https://doi.org/10.5067/8AE2ILXB5SM6.